# Pre-Diabetes-Linked miRNA miR-193b-3p Targets PPARGC1A, Disrupts Metabolic Gene Expression Profile and Increases Lipid Accumulation in Hepatocytes: Relevance for MAFLD

**DOI:** 10.3390/ijms24043875

**Published:** 2023-02-15

**Authors:** Inês Guerra Mollet, Maria Paula Macedo

**Affiliations:** 1iNOVA4Health, NOVA Medical School (NMS), Faculdade de Ciências Médicas (FCM), Universidade NOVA de Lisboa, 1150-082 Lisboa, Portugal; 2UCIBIO-Requimte, Faculdade de Ciências e Tecnologia (FCT), Universidade NOVA de Lisboa, 2825-149 Caparica, Portugal; 3Associação Protectora dos Diabéticos de Portugal, Education Research Center (APDP-ERC), 1250-203 Lisbon, Portugal

**Keywords:** metabolic syndrome, liver, hsa-miR-193b-3p, peroxisome proliferator activated receptor gamma coactivator 1 alpha, lipid metabolism

## Abstract

Distinct plasma microRNA profiles associate with different disease features and could be used to personalize diagnostics. Elevated plasma microRNA hsa-miR-193b-3p has been reported in patients with pre-diabetes where early asymptomatic liver dysmetabolism plays a crucial role. In this study, we propose the hypothesis that elevated plasma hsa-miR-193b-3p conditions hepatocyte metabolic functions contributing to fatty liver disease. We show that hsa-miR-193b-3p specifically targets the mRNA of its predicted target *PPARGC1A*/PGC1α and consistently reduces its expression in both normal and hyperglycemic conditions. *PPARGC1A*/PGC1α is a central co-activator of transcriptional cascades that regulate several interconnected pathways, including mitochondrial function together with glucose and lipid metabolism. Profiling gene expression of a metabolic panel in response to overexpression of microRNA hsa-miR-193b-3p revealed significant changes in the cellular metabolic gene expression profile, including lower expression of *MTTP*, *MLXIPL*/ChREBP, *CD36*, *YWHAZ* and *GPT*, and higher expression of *LDLR*, *ACOX1*, *TRIB1* and *PC*. Overexpression of hsa-miR-193b-3p under hyperglycemia also resulted in excess accumulation of intracellular lipid droplets in HepG2 cells. This study supports further research into potential use of microRNA hsa-miR-193b-3p as a possible clinically relevant plasma biomarker for metabolic-associated fatty liver disease (MAFLD) in dysglycemic context.

## 1. Introduction

MicroRNAs are small, non-protein-coding, 20–22-nucleotide-long RNAs produced by cells of various tissues that can be actively released into the bloodstream in vesicles called exosomes and delivered to cells in the same or other tissues in the organism [1]. MicroRNAs target protein expression essentially by binding to the 3’ untranslated region (3’UTR) of messenger RNA (mRNA), thereby preventing translation of the mRNA into protein and directing the mRNA to nonsense-mediated decay [2]. We hypothesize that particular microRNAs detected in circulation in early disease states can be used as biomarkers of underlying dysmetabolism and also that the targets [3] of each microRNA can provide specific crucial organ-specific pathological information that could be used to inform early clinical intervention. Several circulating miRNAs have been reported as being altered in plasma from patients with metabolic disease, including pre-diabetes; however, the specific mechanisms through which they function remain unexplored [4].

Pre-diabetes, diagnosed as impaired fasting glucose (IFG: only fasting glycemia, between 100 and 125 mg/dL) and/or impaired glucose tolerance (IGT: glycemia 2 h after glucose challenge, between 140 and 199 mg/dL), is considered an early sign of metabolic syndrome [5]. NAFLD/MAFLD (nonalcoholic fatty liver disease/metabolic-associated fatty liver disease) [6] is a precursor of metabolic syndrome [7,8]. NAFLD/MAFLD is a condition characterized by a fatty liver, which can lead to complications, including cirrhosis, liver failure, liver cancer and cardiometabolic health problems that encompass dysmetabolism. Thus, analyzing the effects of pre-diabetes-associated circulating microRNA targets in liver cells is an appropriate premise to identify early intervention targets that may have potentially important clinical application in diagnosis, prevention and treatment, paving the way for precision medicine. Among these is microRNA hsa-miR-193b-3p, which is overexpressed in plasma of patients with impaired fasting glucose and impaired glucose tolerance and which tends to normalize upon exercise intervention in humans [4,9] but for which no data exist on a precise mechanism or relevant targeting.

In this study, we demonstrate that hsa-miR-193b-3p targets the expression of *PPARGC1A* mRNA and increases lipid accumulation in human hepatocyte-derived cells in hyperglycemic condition. *PPARGC1A* codes for PGC-1α, a central co-activator of transcriptional cascades that regulate several interconnected pathways including glucose metabolism [10], lipid metabolism [11,12], inflammation, mitochondrial function and oxidative stress [13]. PGC-1α functions by interacting with the nuclear peroxisome proliferator-activated receptors (PPARs), activating them to bind DNA in complex with retinoid X receptor (RXR) [14] on the promoter of numerous genes central to metabolic regulation, increasing transcription of specific genes and decreasing transcription of others. The peroxisome proliferator activated receptors (PPARs) are nuclear receptors that play key roles in the regulation of lipid and glucose metabolism, inflammation, cellular growth and differentiation. The receptors bind and are activated by a broad range of fatty acids and fatty acid derivatives serving as major transcriptional sensors of fatty acids [14]. In order to determine whether the expression of genes involved in metabolic pathways was also affected by overexpression of hsa-miR-193b-3p, we analyzed the expression of several metabolic master-switch genes, some of which are known to be transcriptionally regulated by *PPRAGC1A*/PGC1α, such as *GPT* [15]. Glucose and lipid metabolism depend on the ability of mitochondria to generate energy in cells; therefore, we also looked at genes involved in hepatic mitochondrial dysfunction, which plays a central role in the pathogenesis of insulin resistance in obesity, pre-diabetes and NAFLD/MAFLD [16]. 

We sought to investigate whether elevated levels of microRNA hsa-miR-193b-3p in hepatocytes could interfere with hepatic cell function in vitro under conditions that mimic pre-diabetes (hyperglycemia and hyperinsulinemia). We evaluated pre-diabetes associated microRNA hsa-miR-193b-3p predicted target *PPARGC1A* (PGC1α), hepatic lipid content and direct or indirect effects on expression of genes regulating cellular energy metabolism, in human hepatocyte-derived cells. The objective was to investigate which metabolic pathways microRNA hsa-miR-193b-3 interfered with so as to gauge the potential of hsa-miR-193b-3p as an early plasma biomarker of specific liver dysmetabolism to support precision medicine in early stages.

## 2. Results

### 2.1. MicroRNA hsa-miR-193b-3p Targets the 3’UTR of PPARGC1A mRNA and Downregulates PPARGC1A Expression in HepG2 Cells

To evaluate potential mRNA targets of microRNA hsa-miR-193b-3p, we performed a bioinformatic analysis of conserved predicted targets of hsa-miR-193b-3p from the online database TargetScan 7.1 [3] using the DAVID Bioinformatics Resources 6.8 [17,18] and clustering tools for functional Gene Ontology Annotations [19]. Of the 283 predicted conserved mRNA targets of microRNA hsa-miR-193b-3p, we found the top gene ontology cluster was composed of 43 genes involved in gene transcription. From this list, *PPRAGC1A* stood out as the gene with the most consistently conserved microRNA hsa-miR-193b-3p target site involved in energy metabolism. Peroxisome proliferator-activated receptor gamma coactivator 1-alpha (*PPARGC1A*) codes for PGC-1α, a transcriptional coactivator of numerous genes involved in energy metabolism. The endogenous level of hsa-miR-193b-3p in HepG2 cells was measured using real-time RT-qPCR. We observed identical Ct values of hsa-miR-193b-3p and the reference snoRNA RNU44 (Ct = 26.8 ± 0.4 and Ct = 26.7 ± 0.3, respectively), while snoRNA RNU48 had higher expression with Ct = 23.7 ± 0.5. Parrizas et al. 2015 [9] showed that higher levels of hsa-miR-193b-3p were observed in plasma of people with impaired fasting glucose by a difference of almost 6 dCt between lowest levels in non-diabetic controls and highest level in impaired fasting glucose, which corresponds to a 64-fold difference. To determine a physiologically relevant overexpression of hsa-miR-193b-3p to use in this study, we measured, by real-time RT-qPCR, the fold increase in hsa-miR-193b-3p using a 0.05 nM, 0.5 nM and 5 nM transfection of hsa-miR-193b-3p against the scrambled control microRNA and observed a fold increase of 1.9, 11.1 and 131, respectively. We therefore selected 5 nM hsa-miR-193b-3p as an appropriate physiologically relevant level of overexpression.

When hsa-miR-193b-3p was overexpressed in HepG2 cells for 72 h using a mimic, we observed robust downregulation of *PPARGC1A* mRNA expression at basal glucose concentration (5 mM glucose, Figure 1A), under hyperglycemia (20 mM glucose, Figure 1B), under hyperinsulinemia (5 mM glucose and final 24 h 10 nM insulin, Figure 1C) and under hyperglycemia/hyperinsulinemia (20 mM glucose and final 24 h 10 nM insulin, Figure 1D). We validated the predicted *PPARGC1A* 3’UTR target site for miR-193b-3p by cloning a “match”, or a “mismatch” control (Figure 1E), into the 3’UTR of the firefly luciferase gene in the pmirGLO dual-luciferase miRNA target expression vector (Figure 1F), followed by dual-glow luciferase assay. Expression of firefly luciferase from the “match” clone was significantly downregulated in HepG2 cells when the hsa-miR-193b-3p mimic was overexpressed and cells were transfected with the pmirGLO match clone (Figure 1G). These results support our posit that microRNA hsa-miR-193b-3p directly targets and downregulates the human *PPARGC1A* mRNA by binding to its 3’UTR.

### 2.2. Intracellular Lipid Droplet Content Is Increased by Overexpression of microRNA hsa-miR-193b-3p in HepG2 Cells

*PPARGC1A*/PGC1α is of central importance for lipid regulation [11]. Fatty liver is associated with impaired activity of *PPARGC1A* [20]. Therefore, having verified that hsa-miR-193b-3p overexpression in HepG2 hepatoma cells reduced the expression of *PPARGC1A*, we proceeded to analyze the intracellular lipid content in these cells. We overexpressed hsa-miR-193b-3p mimic in HepG2 cells for 72 h, under normoglycemia (5 mM glucose) and hyperglycemia (20 mM glucose); visualized lipid droplet content using Oil Red O staining and brightfield microscopy (Figure 2); and performed quantitative image analysis. At hyperglycemia (20 mM glucose) we observed a significant increase in total area covered by lipid droplets (Figure 2G) and number of lipid droplets (Figure 2H); we also saw an increasing trend in lipid droplet size (Figure 2I), though not statistically significant. All three measurements also showed increasing trends at 5 mM. This result supports further research into the possibility of using increased plasma levels of microRNA hsa-miR-193b-3p as a biomarker for fatty liver disease.

### 2.3. Changes in Expression of Genes Involved in Lipid Metabolism May in Part Explain Intracellular Lipid Accumulation Following Overexpression of hsa-miR-193b-3p in HepG2 Cells

Having observed that *PPARGC1A* expression is robustly downregulated and lipid droplets accumulate when microRNA hsa-miR-193b-3p is overexpressed in HepG2 cells under hyperglycemia–hyperinsulinemia, we proceeded to analyze the expression of other genes involved in lipid metabolism and processing that might be affected. The lower mRNA expression of microsomal triglyceride transfer protein (*MTTP*) (Figure 3), which is required for the secretion of plasma lipoproteins containing apolipoprotein B [21], such as VLDL, that is made in hepatocytes and secreted to deliver lipids, may be in part responsible for the increased lipid droplet accumulation observed. The increased expression of Tribbles pseudokinase 1 (*TRIB1*) (Figure 3) could also be responsible for the lipid droplet accumulation we observed [22]; indeed, Burkhardt et al. (2010) reported that hepatic overexpression of *TRIB1* in mice led to reduced VLDL secretion. The increased expression of low-density lipoprotein receptor (*LDLR*) (Figure 3) may increase LDL uptake in favor of increased lipid accumulation. No significant expression changes were observed in apolipoprotein B (*APOB*) (Figure 3).

Expression of carbohydrate-responsive element-binding protein (ChREBP), also known as MLX-interacting protein-like (*MLXIPL*), showed a reduced trend of 22% (vs. mock control) and 32% (vs. negative control) (Figure 3) when hsa-miR-193b-3p was overexpressed. This transcription factor binds and activates carbohydrate response element (ChoRE) motifs on the promoters of triglyceride synthesis genes in a glucose-dependent manner. Thus, we expect to see reduced expression of genes involved in triglyceride synthesis such as *DGAT2* [23], which catalyzes the final reaction in triglyceride synthesis. However, no changes in expression of *DGAT2* (Figure 3) were observed. No expression changes were observed in either of two major transcriptional regulators, sterol regulatory element binding transcription factors 1 and 2 coded by *SREBF1* (Figure 3) and *SREBF2* (Figure 3), which, in concert with *MLXIPL*/ChREBP, regulate hepatic lipid metabolism by inducing transcription of lipogenic enzymes that direct lipogenesis in the liver [24]. We also observed a 20% reduction in expression of fatty acid synthase (*FASN*) (vs. mock) (Figure 3), indicating reduced fatty acid synthesis, which was expected since this gene is regulated by *MLXIPL*/ChREBP. However, no change was observed in acetyl-CoA carboxylase (*ACACA*) (Figure 3), the enzyme which catalyzes the carboxylation of acetyl-CoA to malonyl-CoA, the rate-limiting step in fatty acid synthesis. This lack of downregulation of *ACACA* is important, because although we observe increased expression of Acyl-CoA oxidase 1 (*ACOX1*) (Figure 3), indicating increased fatty acid oxidation in the mitochondria, the malonyl-CoA produced by *ACACA* in the cytoplasm can inhibit transfer of cytoplasmic fatty acids into the mitochondria for beta oxidation contributing to increased lipids in the cytoplasm. Unexpectedly, we observed a 50% reduction in the fatty acid translocase *CD36* (Figure 3), which has been described as positively correlated with fatty liver [25]. Thus, the changes we observed in mRNA levels of genes involved in lipid metabolism do not all provide a clear rationale to explain lipid accumulation in hepatocytes during overexpression of hsa-miR-139b-3p.

### 2.4. Changes Favoring Insulin Signaling, Reduced Glycolysis, and Reduced Mitochondrial Biogenesis Are Observed When microRNA hsa-miR-193b-3p Is Overexpressed in HepG2 Cells

In addition to lipid metabolism, *PPARGC1A*/PGC1α also regulates insulin signaling, hepatic glucose metabolism and mitochondrial turnover [20,26,27]. Therefore, we investigated whether expression of genes involved in these mechanisms were altered. We observed downregulation of *YWHAZ* (Figure 4). *YWHAZ* belongs to the 14-3-3 family of proteins that mediate signal transduction; it binds to insulin receptor substrate-1 (IRS-1) and is thought to interrupt the association between the insulin receptor and IRS-1, thus interfering with insulin signaling [28]. Our result indicates decreased interference with insulin signaling by this particular mechanism. No change was observed in Tribbles homolog 3 (*TRIB3*) (Figure 4) that has been linked to insulin resistance and hepatic production of glucose in the liver [29,30]. 

*PPARGC1A*/PGC1α is known to regulate key hepatic gluconeogenic enzymes leading to increased glucose output [27], and we saw a significant increase in expression of pyruvate carboxylase (*PC*) (Figure 4), the mitochondrial matrix enzyme that regulates fuel partitioning toward gluconeogenesis in hepatocytes. We saw reduced expression trend of the plasma membrane bidirectional glucose transporter *SLC2A2*/GLUT2 (Figure 4), indicating both reduced uptake and release of glucose. The reduced expression of liver phosphofructokinase (*PFKL*) (Figure 4) and pyruvate kinase (*PKLR*) (Figure 4) indicate reduced glycolysis. Reduced expression of the pyruvate dehydrogenase E1 subunit alpha 1 (*PDHA1*) (Figure 4), a central component of the pyruvate dehydrogenase complex that is the primary link between glycolysis and the TCA cycle, indicated lower conversion of pyruvate to acetyl-CoA in mitochondria. This is further supported by a five-fold increased expression of the pyruvate dehydrogenase inhibitor mitochondrial kinase *PDK4* (Figure 4). Reduced expression of glutamic-pyruvic transaminase *GPT* (Figure 4) suggests lower conversion of alanine to pyruvate. Based on all this, we can predict a drop in ATP production.

We also observed reduced expression of mitochondrial transcription factor A (*TFAM*) (Figure 4) that is required for mitochondrial biogenesis through maintenance of mitochondrial DNA and replication of normal levels of mitochondrial DNA; reduced expression trend of mitofusin-2, *MFN2* (Figure 4), which regulates mitochondrial fusion dynamics; and reduced expression trend of isocitrate dehydrogenase 3 catalytic subunit alpha (*IDH3A*) (Figure 4), which catalyzes the rate-limiting step of the TCA cycle in the mitochondrial matrix.

## 3. Discussion

This study is based on recent research showing that circulating microRNAs detected in plasma can enter cells in various tissues and there interfere with gene expression [31], and that, in humans, altered levels of circulating microRNAs are detected in many disease states including metabolic diseases [4]. In humans, higher than normal levels of the microRNA hsa-miR-193b-3p have been detected in plasma of patients with impaired fasting glucose and impaired glucose tolerance, two parameters that are used to diagnose pre-diabetes, an early sign of metabolic syndrome [5]. Nonalcoholic fatty liver disease, also referred to as metabolic-associated fatty liver disease (NAFLD/MAFLD) [6,7,8], characterized by excess accumulation of lipids in the liver (known as steatosis) is considered a precursor of metabolic syndrome. In this study, we have focused on identifying and validating a direct target of hsa-miR-193b-3p and evaluating changes in expression of genes involved in cellular energy metabolism after overexpression of microRNA hsa-miR-193b-3p in the hepatocyte cell line HepG2, which is considered an appropriate cell model [32]. This strategy is in line with recent attempts to try to understand metabolism in type 2 diabetes beyond glycemia [33]. 

Our results show that microRNA hsa-miR-193b-3p directly targets *PPRAGC1A*/PGC1α, reducing its expression, and that it causes excess intracellular lipid droplet accumulation in HepG2 cells. We also show that the accumulation of lipids observed may be caused in part by altered expression of genes involved in lipid processing, which may cause reduced VLDL secretion, as well as reduced glycolysis and reduced mitochondrial activity shifting the metabolic balance toward lipogenesis. We observed a 60% reduction in *MTTP* required for VLDL secretion, and a 3-fold increased expression of *TRIB1*, which reduces secretion of VLDL. Indeed, hepatic overexpression of Trib1 in mice reduces secretion of VLDL from the liver into circulation [34]. Although no change in *APOB* was observed, the decreased expression trend of *MLXIPL*/ChREBP suggests a reduced rate of triglyceride synthesis that could lead to insufficient *APOB* lipidation and secretion.

Insulin signaling normally directs de novo lipogenesis in the liver [35], among several other pathways. The observed downregulation of *YWHAZ*, known to bind insulin receptor substrate-1 (IRS-1) [28], predicts reduced interference of *YWHAZ* with IRS-1 at this point in the insulin signaling cascade. However, linking this observation to increased lipogenesis requires investigating additional downstream steps in the insulin signaling cascade towards lipogenesis. 

When insulin and glucose are associated in liver cells, as in our experimental conditions, the stimulatory effect of glucose on *SLC2A2*/GLUT2 gene expression is predominant [36]. However, when hsa-miR-193b-3p was overexpressed, we observed a decreasing trend in *SLC2A2*/GLUT2 expression, suggesting lower glucose uptake, which may contribute to hyperglycemia, a feature associated with the plasma increase in hsa-miR-193b-3p in humans. We also observed lower glycolysis (reduced *PFKL* and *PKLR*), producing less pyruvate; less pyruvate is also expected coming from amino acid metabolism given the reduced *GPT* expression. Inhibition of conversion of pyruvate to acetyl-CoA for the TCA cycle is predicted via reduced *PDHA1* and increased *PDK4* expression; all of this points to a decrease in energy combustion and lower mitochondrial-generated ATP. In addition, we observed lower expression of *TFAM* pointing to less mitochondrial biogenesis, expected to contribute to a decrease in energy combustion. In line with our results, mitochondrial dysfunction has been shown to play a central role in the pathogenesis of metabolic diseases and associated complications [37], and reduced energy combustion through reduced glycolysis has been described as critical to excess lipid storage in the liver [38]. Energy combustion in the liver is modulated by PPARα-regulated fatty acid beta-oxidation. Interference with PPARα lipid sensing can reduce energy burning and result in accumulation of lipids in hepatocytes. Fatty liver is also associated with impaired activity of *PPARGC1A* (PGC1α) and reduced mitochondrial biogenesis in mice [20]. Although we saw increased expression of *ACOX1*, indicating increased fatty acid beta-oxidation in mitochondria, this could be a compensation for reduced mitochondrial biogenesis. We saw no change in acetyl-CoA carboxylase (*ACACA*), the enzyme which catalyzes the carboxylation of acetyl-CoA to malonyl-CoA, the rate-limiting step in fatty acid synthesis; this is relevant, as the malonyl-CoA produced by ongoing fatty acid synthesis will inhibit fatty acid transport into mitochondria for beta-oxidation.

Hepatic disruption of fatty acid translocase *CD36* in JAK2L livers has been shown to lower triglyceride (TG), diacylglycerol (DAG) and cholesterol ester (CE) content, significantly improving steatosis [25]. Therefore, one might speculate that our unexpected observation of a 50% reduction in the fatty acid translocase *CD36* alongside lipid accumulation in HepG2 cells under hyperglycemia/hyperinsulinemia might be a compensatory mechanism.

*PPRAGC1A*/PGC-1α interacts with the PPAR nuclear receptors [14], on the promoter of numerous genes involved in metabolic regulation. However, the gene expression changes that we observed when microRNA hsa-miR-193b-3p was overexpressed under hyperglycemia/hyperinsulinemia in the HepG2 cell model may result from either direct transcriptional regulation via *PPRAGC1A*/PGC-1α or other indirect regulation. In future work, one might consider rescuing the downregulation of *PPRAGC1A*/PGC-1α observed by overexpressing *PPRAGC1A*/PGC-1α along with hsa-miR-193b-3p in this cell model. As there are a further 283 predicted conserved mRNA targets of microRNA hsa-miR-193b-3p, including 43 genes involved in gene transcription, other predicted gene targets of hsa-miR-193b-3p must also be investigated. With regards to the specific targeting of *PPARGC1A*/PGC-1α by hsa-miR-193b-3p, and given the central importance of PGC1α in inflammation, oxidative stress and energy in cells other than hepatocytes, much work remains to be done on these pathways and on other cell types and in vivo. Future work on this subject should also include investigating the effects of inhibiting hsa-miR-193b-3p on hepatocyte lipid content.

Given that *PPRAGC1A*/PGC-1α interacts with the nuclear receptor PPAR-γ [14], our results also warrant bearing in mind the drug class of thiazolidinediones, which are potent PPAR-γ agonists, that can be used in the treatment of type 2 diabetes to improve hepatic sensitivity to insulin [39,40] and improve fibrosis in nonalcoholic steatohepatitis [40,41]. In this context, further work should differentiate effects which are mediated directly through hsa-miR-193b-3p targeting of *PPRAGC1A*/PGC-1α and PPAR transcriptional regulation, and which might be indirect.

Our results show that elevated levels of microRNA hsa-miR-193b-3p in HepG2 hepatocytes cause a cascade of gene-expression changes and increased lipid accumulation in these cells, and specific direct targeting of *PPARGC1A*/PGC1α mRNA. As levels of plasma hsa-miR-193b-3p may be elevated in pre-diabetes in humans [9], these results support the hypothesis of using hsa-miR-193b-3p as an early diagnostic biomarker of liver dysmetabolism in pre-diabetes. The potential clinical relevance of these results is highlighted by recent research showing that improvement of fatty liver disease reduces the risk of type 2 diabetes [42]. Thus, plasma levels of hsa-miR-193b-3p could conceivably be used to flag pre-diabetic patients for early intervention to target asymptomatic fatty liver, thereby preventing aggravation of pre-diabetes and the development of type 2 diabetes.

## 4. Materials and Methods

Cell model. The HegG2 hepatoma cell line [32] was maintained in Dulbecco’s modified Eagle’s medium (DMEM, 25 mM high glucose (21969-035, Gibco, Life Technologies, Carlsbad, CA, USA), 4 mM glutamine (2503-149, Gibco, Life Technologies, Carlsbad, CA, USA), 10% heat inactivated fetal bovine serum (S0615, Biochrom, Cambridge, UK), 100 IU/mL penicillin and 100 ug/mL streptomycin (15140-122, Life Technologies, Carlsbad, CA, USA. Experiments at basal 5 mM glucose were performed with DMEM 4 mM glutamine, 1 mM sodium pyruvate (SH30021.FS, Thermo Scientific), 10% heat-inactivated fetal bovine serum, 100 IU/mL penicillin and 100 ug/mL streptomycin.

Overexpression of miR-193b-3p in HepG2 cells. microRNA miR-193b-3p overexpression in HepG2 cells was performed over 72 h by reverse transfection on the day of plating, followed by forward transfection the next day, with an additional day of growth, before RNA extraction, plasmid transfection or fixing cells for imaging. MicroRNA mimics used were hsa-miR-193b-3p miRCURY LNA™ microRNA Mimic (472852-001, Exiqon, Hovedstaden, Denmark) and Negative Control 4 miRCURY LNA™ microRNA Mimic (479903-001, Exiqon, Hovedstaden, Denmark). Transfection of microRNA mimic and control into HegG2 cells was performed at 5 nM final concentration using Lipofectamine^®^ RNAiMAX Transfection Reagent (13778-100, Life Technologies, Carlsbad, CA, USA) with 20 min lipofectamine/mimic complex formation in Opti-MEM^®^ I Reduced Serum Medium (31985-062, Life Technologies, Carlsbad, CA, USA) and cells incubated in DMEM culture as described above with no penicillin or streptomycin added. Mock transfection was performed with lipofectamine alone.

RNA preparation and relative gene expression. Total RNA was prepared by organic extraction using Trizol Reagent (15596-018, Life Technologies, Carlsbad, CA, USA) and chloroform (Sigma, St Louis, MO, USA). RNA was precipitated in absolute ethanol, and dissolved in DNase/RNaseFree Distilled Sterile Water (10977-035, Gibco, Billings, MT, USA). RNA concentration was determined using Nanodrop 2000 (Thermo Fisher Scientific, Waltham, MA, USA). Relative gene expression was determined by two-step reverse-transcription real-time quantitative PCR (real-time RT-qPCR) with cDNA prepared from total extracted RNA using a High-Capacity cDNA Reverse Transcription Kit (4368814, Applied Biosystems, Waltham, MA, USA) on a PCR MyCycler (Biorad, CA, USA) (25 °C 10 min, 37 °C 120 min, 85 °C 5 min). For microRNA expression, the protocol was modified according to the manufacturer’s instructions with added RNAse Inhibitor (N8080119, Life Technologies, Carlsbad, CA, USA) and pooled stem loop primer mix of hsa-miR-193b-3p, RNU44 and RNU48 (0.3 uL each stem-loop primer in 15 uL reaction volume: stem loop primers (TaqMan^®^ MicroRNA Assay, hsa-miR-193b-3p, Assay ID 002366, 002366/4427975, Life Technologies, Carlsbad, CA, USA). The two internal reference genes used were TaqMan^®^ MicroRNA Control Assays RNU44 (Assay ID 001094, 001094/4427975, Life Technologies, Carlsbad, CA, USA) and RNU48 (Assay ID 001006, 001006/4427975, Life Technologies, Carlsbad, CA, USA). TaqMan Universal Master Mix II, with UNG (4440042, Life Technologies, Carlsbad, CA, USA), was used to determine relative expression of the miRNA miR-193b-3p using TaqMan MicroRNA Assays qPCR primers; the geometric mean of both references was used to calculate 2^−ddCt.^ For protein-coding genes, forward and reverse primers for qPCR for each gene of interest were designed across constitutive exons using Primer Quest (Integrated DNA Technologies, IDT, https://www.idtdna.com/pages/tools/primerquest, accessed on 24 September 2018). The list of primers used is presented in Table 1. The custom-designed primers were ordered from Invitrogen (Waltham, MA, USA). Relative gene expression was determined from cDNA using NZYTaq 2× Green Master Mix (MB03903, NZYtech, Lisbon, Portugal) on MicroAmp Optical 96-well reaction plates (N8010560, Applied Biosystems, Waltham, MA, USA). Amplification and Ct values for two technical replicates of each sample were obtained on qPCR LightCycler and integrated software (Roche, Boston, MA, USA). Amplicon specificity was verified by a high-resolution melting curve using LightCycler integrated software. Relative mRNA expression was determined using the relative gene expression data using real-time quantitative PCR and the 2^−ddCt^ method [43] using *TBP* primers as reference gene.

miRNA target plasmid construct and luciferase assay. To validate the miR-193 b-3p target site on the mRNA of *PPARGC1A* (PGC1a gene), the pmirGLO dual-luciferase miRNA target expression vector (E1330, Promega, Madison, WI, USA) was used to clone the predicted *PPARGC1A* target site for microRNA miR-193b-3p along with a mismatch control (Figure 1E,F). The custom primers for cloning were ordered from Invitrogen. Primer pairs were cloned by double digestion using *Sac*I and *Sal*I restriction enzymes (R0156, R0156, CutSmart Buffer, New England Biolabs, Ipswich, MA, USA) and ligation with T4 DNA Ligase (EL0014, Thermo Fisher Scientific, Waltham, MA, USA) and FastAP thermosensitive alkaline phosphatase (EF0654, Thermo Fisher Scientific, Waltham, MA, USA). Selection of clones was made with *Xho*I/*Bam*HI double digest (R0136, R0146, CutSmart Buffer, New England Biolabs, Ipswich, MA, USA), with the *Xho*I site being on the excised multiple cloning site and *Bam*HI being present on the core plasmid. Digestion of positive clones produces a single fragment as opposed to two fragments. Clones were visualized after electrophoresis on 2% agarose using GreenSafe Premium (MB13201, NZYTech, Lisbon, Portugal) on Chemidoc Touch (Biorad, CA, USA). Plasmids were reproduced by chemical transformation and culture of DH5alpha *E. coli* (inhouse stock) in LB medium (1% tryptone, 0.5% yeast extract, 1% NaCl, pH 7.0). Colonies were grown on 1.5% agar/LB (20767.232, VWR Chemicals, Radnor, PA, USA) with ampicillin selection (20767.232, VWR Chemicals, Radnor, PA, USA). Plasmid DNA was extracted and purified using a Plasmid DNA Mini Kit I (D6943-01, E.Z.N.A., Norcross, GA, USA). Plasmid clones were sequenced to verify correct cloned insert with custom sequencing primer 5’-CGAACTGGAGAGCATCCTG-3’ using StabVida SANGER sequencing services (https://www.stabvida.com/sanger-sequencing-service, accessed on 17 January 2020). The selected pmirGLO dual-luciferase plasmid clone containing the hsa-miR-193b-3p target site from *PPARGC1A* was transfected into HepG2 cells grown at 5 mM glucose using the DharmaFECT kb DNA transfection reagent lipofectamine (T-2006-01, Dharmacon, Lafayette, CO, USA). Expression of firefly luciferase was normalized to internal renilla luciferase using a Dual-Glo^®^ Luciferase Assay System (E2920, Promega, Madison, WI, USA), with chemiluminescent data obtained in relative light units (RLU) measured using SpectraMax i3x (Molecular Devices, San Jose, CA, USA).

Lipid droplet quantification. To identify intracellular lipid content, HepG2 cells were cultured at 5 mM glucose or 20 mM glucose on 13 mm glass coverslips in 24-well plates. At the end of the experiment, cells were fixed with 4% paraformaldehyde in phosphate-buffered saline (PBS) at room temperature for 15 min, then washed with PBS before staining with the Oil Red O method [44] using a kit (010303, Diapath, Italy) according to the manufacturer’s instructions. Nuclei were stained blue with hematoxylin. Coverslips were mounted on glass slides for microscopy. Stained cells were analyzed on a Zeiss Z2 fluorescent microscope with ZEN Pro 2012 software (Zeiss, Oberkochen, Germany. For each condition, ten brightfield color images were captured at 40× magnification with an Axiocam 105 color camera. Image quantitative analysis of lipid droplets was performed on each of ten images per condition with Fiji (ImageJ) software [45] using Color Deconvolution (v1.7, FastRed FastBlue DAB plugin); red color was used for binary/watershed particle analysis to obtain parameters of total red area (pixels^2^), average particle size (pixels) and number of particles; blue color was used for manual nuclei counting using Fiji software; number of nuclei per image was used for normalization of red data per image. Results per condition are the average of ten images.

Statistical Analysis. Statistical analyses were performed using Excel functions. Data are presented as mean ± standard error of the mean (SEM). Statistical significance was evaluated using two-tailed unpaired two-sample equal variance Student’s *t*-test, with 0.05 threshold chosen for statistical significance; *p*-values indicated by # < 0.1, * < 0.05, ** < 0.01, *** < 0.001 and **** < 0.0001.

## Figures and Tables

**Figure 1 ijms-24-03875-f001:**
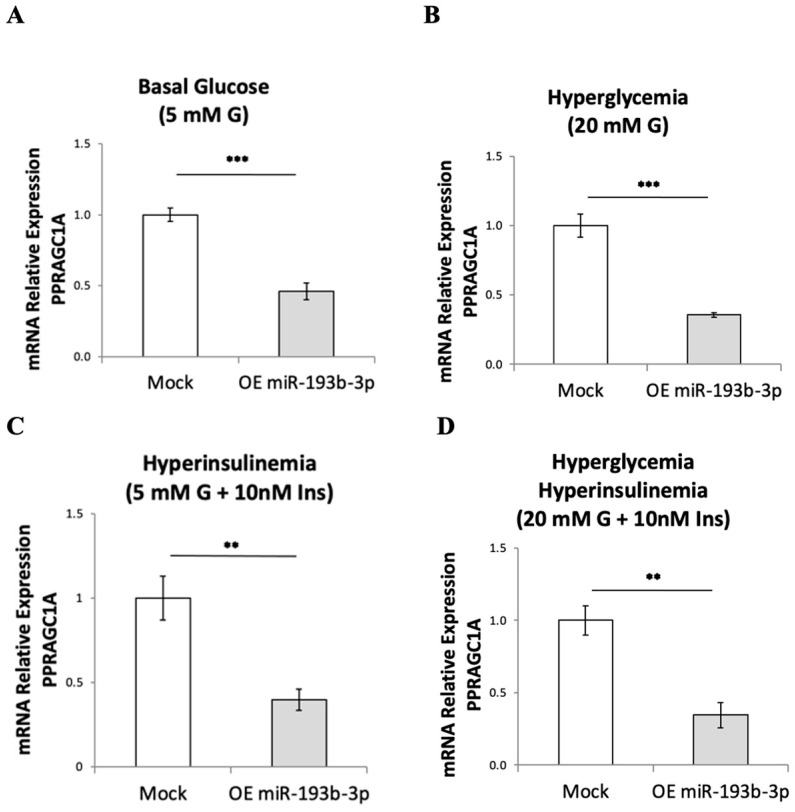
MicroRNA hsa-miR-193b-3p targets the 3’UTR of *PPARGC1A* mRNA and downregulates *PPARGC1A* expression in HepG2 cells. (**A**–**D**) mRNA expression, evaluated by real-time RT-qPCR, of *PPARGC1A* after a 72 h overexpression of hsa-miR-193b-3p in HepG2 cells in basal 5 mM glucose (**A**); hyperglycemia at 20 mM glucose (**B**); hyperinsulinemia with 5 mM glucose and final 24 h with 10 nM insulin (**C**); and hyperglycemia/hyperinsulinemia at 20 mM glucose and final 24 h with 10 nM insulin (**D**). (**E**) Match and mismatch primer pairs containing the *PPARGC1A* 3’UTR target site for hsa-miR-193b-3p that were cloned into pmirGLO miRNA target expression vector (**F**). (**G**) Luciferase luminescence measurement after overexpression of hsa-miR-193b-3p in HepG2 cells; pmirGLO plasmid was transfected into HepG2 cells 24 h before luciferase assay. Data are presented as mean ± SEM of n = 4. Mock—mock transfection, OE—overexpression of hsa-miR-193b-3p, G—glucose, Ins—insulin. Student’s *t*-test *p*-values indicated by * < 0.05, ** < 0.01, *** < 0.001.

**Figure 2 ijms-24-03875-f002:**
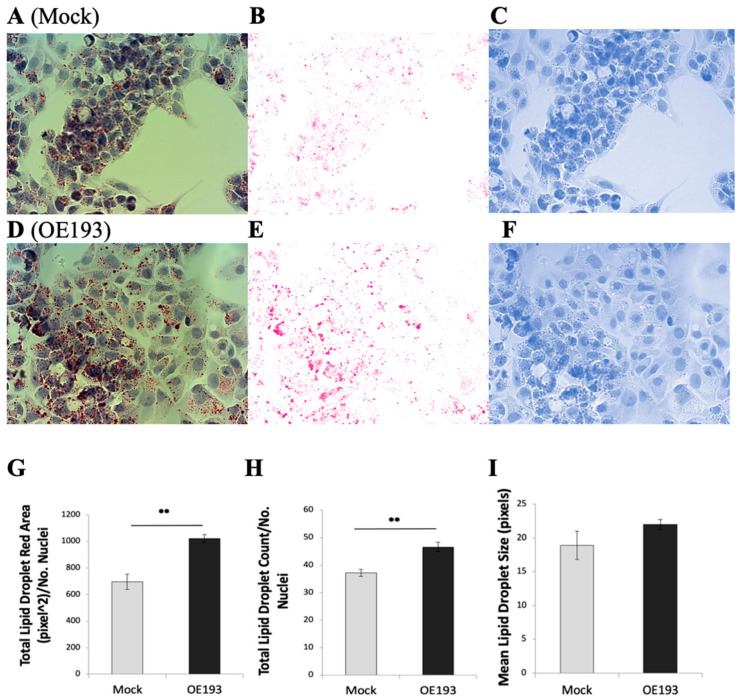
MicroRNA hsa-miR-193b-3p overexpression in HepG2 cells increases intracellular lipid droplet accumulation. HepG2 cells cultured at 20 mM glucose. Lipid droplets in HepG2 cells were visualized using Oil Red O staining. Brightfield 40× magnification images of Oil Red O staining in control mock transfection (**A**) with corresponding red (**B**) and blue (**C**) color deconvolution. Brightfield 40× magnification images of Oil Red O staining after microRNA hsa-miR193b-3p overexpression (**D**) with corresponding red (**E**) and blue (**F**) color deconvolution. Red color deconvolution used for red particle analysis; blue color deconvolution used for nuclei counting. Quantitative analysis of lipid droplet area (**G**), lipid droplet number (**H**) and lipid droplet size (**I**). Data are presented as mean ± SEM of n = 4; each n is average of ten fields per condition normalized to number of nuclei. Mock—mock transfection, OE—overexpression of hsa-miR-193b-3p. Student’s *t*-test *p*-value ** < 0.01.

**Figure 3 ijms-24-03875-f003:**
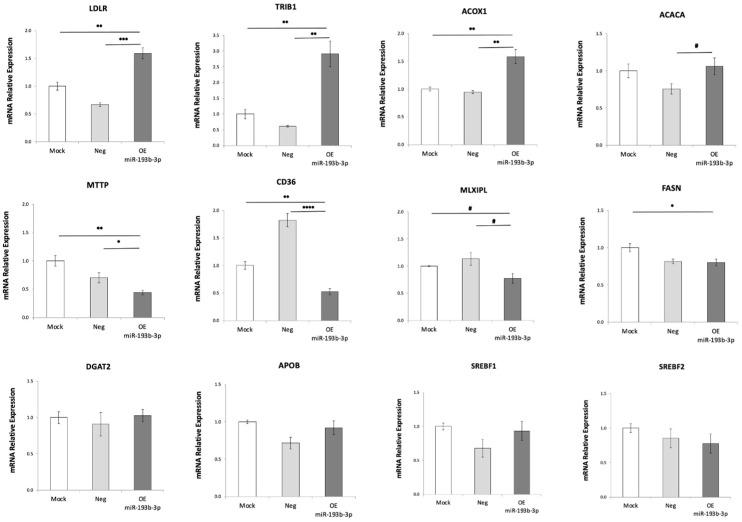
Overexpression of microRNA hsa-miR-193b-3p in HepG2 alters mRNA expression of fundamental genes involved in lipid processing. Lower mRNA expression of *MTTP* and higher expression of *TRIB1* suggest reduced VLDL secretion. Higher level of LDL receptor *LDLR*. No change in *APOB*. Lower mRNA level trend of *MLXIPL*/ChREBP, coordinating triglyceride synthesis. No change in *DGAT2*, *SREBF1* (n = 3) or *SREBF2* (n = 3). Lower expression of fatty acid synthase (*FASN*) (I). No change in *ACACA*. Increased *ACOX1* indicates increased fatty acid oxidation. Reduced fatty acid translocase *CD36*. Relative mRNA expression was evaluated by real-time RT-qPCR after 72 h overexpression of hsa-miR-193b-3p in HepG2 cells under hyperglycemia/hyperinsulinemia (20 mM glucose, with 10 nM insulin during last 24 h). Data are presented as mean ± SEM of n = 4, unless otherwise indicated. Mock—mock transfection, OE—overexpression of hsa-miR-193b-3p. Student’s *t*-test *p*-values indicated by # < 0.1, * < 0.05, ** < 0.01, *** < 0.001 or **** < 0.0001.

**Figure 4 ijms-24-03875-f004:**
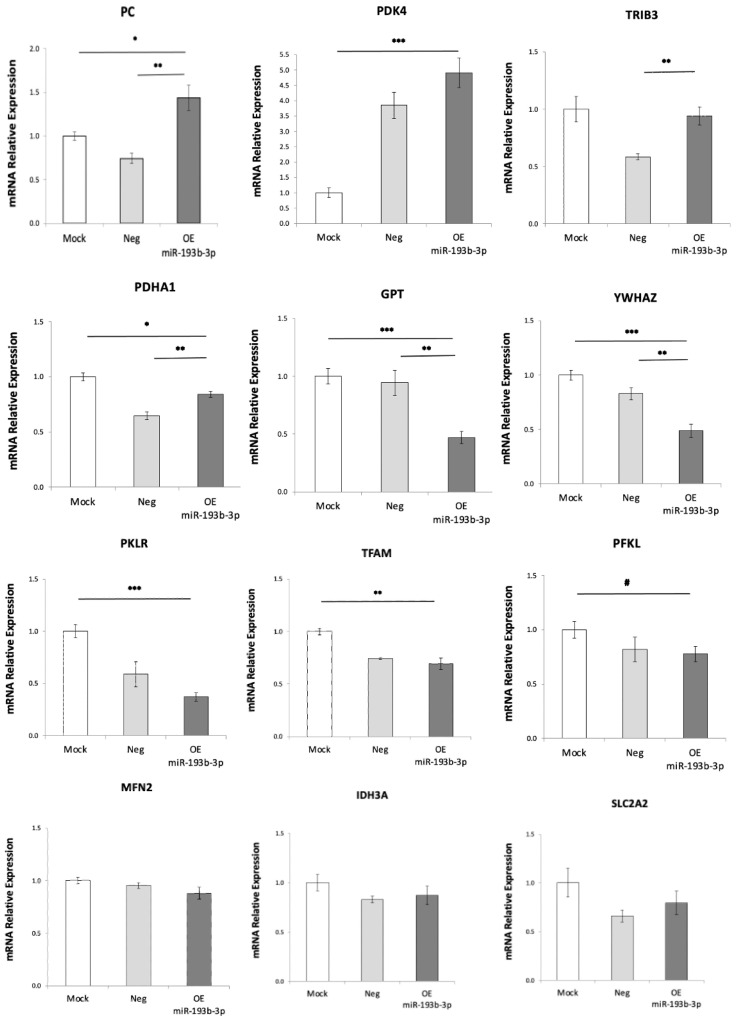
Changes in expression of genes regulating insulin signaling, glucose metabolism, pyruvate availability and mitochondrial biogenesis when hsa-miR-193b-3p is overexpressed in HepG2 cells. Downregulation of *YWHAZ* indicates increased insulin signaling. Increased pyruvate carboxylase (*PC*). Reduced trend in glucose transporter *SLC2A2*/GLUT2 indicates reduced glucose transport. Reduced levels of phosphofructokinase (*PFKL*) and pyruvate kinase (*PKLR*) indicate reduced glycolysis. Reduced levels of pyruvate dehydrogenase *PDHA1* and increase in its inhibiting kinase *PDK4* indicate lower conversion of pyruvate to acetyl-CoA. Reduced glutamic–pyruvic transaminase *GPT* indicates lower conversion of alanine to pyruvate. Reduced levels of mitochondrial transcription factor A (*TFAM*), and reduced trend in mitofusin 2 (*MFN2*) and isocitrate dehydrogenase 3 catalytic subunit alpha (*IDH3A*) indicate reduced levels of mitochondrial biogenesis, fusing dynamics and function, respectively. Relative mRNA expression was evaluated by real-time RT-qPCR after 72 h overexpression of hsa-miR-193b-3p in HepG2 cells under hyperglycemia/hyperinsulinemia (20 mM glucose, with 10 nM insulin during last 24 h). Data are presented as mean ± SEM of n = 4. Mock—mock transfection, OE—overexpression of hsa-miR-193b-3p. Student’s *t*-test *p*-values indicated by # < 0.1, * < 0.05, ** < 0.01, *** < 0.001.

**Table 1 ijms-24-03875-t001:** Primers designed for real-time RT-qPCR.

Gene (Exon Boundary)	Forward Primer	Reverse Primer
*ACACA* (31_32)	TTTGTCAGGATCTTTGATGAAGTG	TCATAAAGAGACGTGTGACCTG
*ACOX1* (13_14)	GATGTGACACTTGGCTCTGT	TTCGTGGACCTCTGCTTTG
*APOB* (20_21)	TTCTGTCAGCGCAACCTATG	CTTCGCACCTTCTGCTTGA
*CD36* (4_5)	GTCCTTATACGTACAGAGTTCGTT	CAGCCTCTGTTCCAACTGATAG
*DGAT2* (6_7)	CAAAGAATGGGAGTGGCAATG	CAGGTCAGCTCCATGACG
*FASN* (3_4)	GTGGACGGAGGCATCAAC	TGTAGCCCACGAGTGTCT
*GPT* (4_5)	CCATCGTGACGGTGCTG	GCCGAGTAGAGTGGGTACT
*IDH3A* (10_11)	GCTCAGTGCCGTGATGAT	GTCAAGCTCTTTCCGTCCTT
*LDLR* (15_16)	CGTAAGGACACAGCACACA	GCCCAGAGCTTGGTGAG
*MFN2* (17_18)	AAGTCCAGCAGGAACTGTC	ATTTCCTGCTCCAGGTTCTC
*MLXIPL* (15_16)	TTTGACCAGATGCGAGACAT	GATGCTGAACACCCAGAACT
*MTTP* (16_17)	CATTCTCAGGAACTTCAGTTACAATC	ACTCACGATACCACAAGCTAAA
*PC* (12_13)	CTGTGGACACCCAGTTCATC	TGACATGGCCGAGGTAGT
*PDHA1* (2_3)	GAAATTAAGAAATGTGACCTTCACC	CAGTCTGCATCATCCTGTAGTA
*PDK4* (7_8)	TCCAGACCAACCAATTCACATC	GCCCGCATTGCATTCTTAAATAG
*PFKL* (13_14)	ATCTCCCATGGACACACAGTAT	TACTTCTTGCACCTGACCCT
*PKLR* (10_11)	CTTTACCGTGAACCTCCAGAAG	CACGGAGCTTTCCACTTTCA
*PPARGC1A* (8_9)	GCAGTAGATCCTCTTCAAGATCC	AACGTGATCTCACATACAAGGG
*SLC2A2* (8_9)	GACGGCTGGTATCAGCAAA	CTCCACAAGGAATACAGAGACAG
*SREBF1* (5_6)	CACTGAGGCAAAGCTGAATAAAT	TAGGTTCTCCTGCTTGAGTTTC
*SREBF2* (2_3)	CTGCAACAACAGACGGTAATG	GCTGAAGGACTTGAAAGCTAGTA
*TBP* (4_5) Reference	TCCACAGTGAATCTTGGTTGT	AGCAAACCGCTTGGGATTA
*TFAM* (2_3_4)	GCTCAGAACCCAGATGCAAA	TGCCACTCCGCCCTATAA
*TRIB1* (2_3)	AGGAGAGAACCCAGCTTAGA	TGGGCAGCCATGTTTGT
*TRIB3* (3_4)	GACCGTGAGAGGAAGAAGC	CTTGTCCCACAGGGAATCAT
*YWHAZ* (5_6)	GAAGCCATTGCTGAACTTGATAC	TCCACAATGTCAAGTTGTCTCT

## Data Availability

All data are contained within the article. All raw data pertaining to the manuscript can be shared upon request to the corresponding author.

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
