# Peer review of "Pre-Diabetes-Linked miRNA miR-193b-3p Targets PPARGC1A, Disrupts Metabolic Gene Expression Profile and Increases Lipid Accumulation in Hepatocytes: Relevance for MAFLD"

_ijms, 2023, doi:10.3390/ijms24043875_

Round 1
Reviewer 1 Report
Pre-diabetes linked miRNA PPARGC1A/PGC1 targets PPARGC1A, 2 disrupts metabolic gene expression profile and increases lipid accumulations in hepatocytes: relevance for MAFLD.
The authors have stated as the hsa-miR-193b-3p specifically 18 targets the mRNA of its predicted target PPARGC1A/PGC1 and consistently reduce its expression 19 in both normal and hyperglycemic condition in line numbers 17-19. There is generally a reduction in PPARGC1A/PGC1 expression both under normal and glycemic conditions. Then, how authors claim overexpression of PPARGC1A/PGC1 is responsible for the downregulation of PPARGC1A/PGC1. Please verify it
We therefore also analyzed the expression of several metabolic master-switch genes downstream of hsa-miR-193b-3p downregulation of PPRAGC1A/PGC1 to analyze downstream pathway regulation. These lines don't convey a clear concept. The sentence needs to be rewritten.
Authors have stated as overexpression of hsa-miR-193b-3p in hepatocytes could reduce the expression of PPARGC1A mRNA. Have the authors analyzed the PPARGC1A mRNA level when inhibiting hsa-miR-193b-3p expression?
Expression of carbohydrate-responsive element-binding protein (ChREBP) also 181 known as MLX-interacting protein-like, MLXIPL, was reduced by 30% (Figure 3E) when hsa-miR-193b-3p was overexpressed. This transcription factor binds and activates carbohydrate response element (ChoRE). The MLXIPL level was reduced when HSA-miR-193b-3p was overexpressed, which indicates that TG synthesis was inhibited. Then how do hepatocytes increase their lipid content?
We also observed a 20% reduction in expression of Fatty Acid Synthase, FASN (Figure 3I) indicating reduced fatty acid synthesis, which was expected since this gene is regulated by ChREBP/MLXIPL. However, no change was observed in Acetyl-CoA carboxylase, ACACA. The same issues were noted when it came to the expression of FAS and ACC, which are both critical enzymes involved in the process of synthesizing fatty acids. The lipid accumulation status in hepatocytes during overexpression of HSA-miR-193b-3p is also controversial.
If the authors could analyze hsa-miR-193b-3p's role while it's inhibited, it would be greatly appreciated.
There is a need to improve the manuscript's language. It is better to avoid using would in several places.
Author Response
Response to Reviewer 1
We thank you in advance and appreciate your comments.
Pre-diabetes linked miRNA PPARGC1A/PGC1 targets PPARGC1A, 2 disrupts metabolic gene expression profile and increases lipid accumulations in hepatocytes: relevance for MAFLD.
The authors have stated as the hsa-miR-193b-3p specifically 18 targets the mRNA of its predicted target PPARGC1A/PGC1 and consistently reduce its expression 19 in both normal and hyperglycemic condition in line numbers 17-19. There is generally a reduction in PPARGC1A/PGC1 expression both under normal and glycemic conditions. Then, how authors claim overexpression of PPARGC1A/PGC1 is responsible for the downregulation of PPARGC1A/PGC1. Please verify it
Response: In this study we did not perform any overexpression of PPARGC1A/PGC1. Nevertheless, our study shows that, in the hepatocyte cell line HepG2, overexpression of hsa-miR-193b-3p consistently lowers mRNA levels of PPARGC1A/PGC1, which was identified as a predicted target of hsa-miR-193b-3p. microRNAs such as hsa-miR-193b-3p lower mRNA levels of their targets by binding to the 3' untranslated region of the mRNA and directing it to nonsense mediated decay.
We therefore also analyzed the expression of several metabolic master-switch genes downstream of hsa-miR-193b-3p downregulation of PPRAGC1A/PGC1 to analyze downstream pathway regulation. These lines don't convey a clear concept. The sentence needs to be rewritten.
Response: Thank you very much for identifying a less clear sentence, consequently we have re-written this sentence to clarify our objectives. It now reads as follows: In order to determine whether the expression of genes involved in metabolic pathways was also affected by overexpression of hsa-miR-193b-3p, we analysed the expression of several metabolic master-switch genes, some of which are known to be transcriptionally regulated by PPRAGC1A/PGC1a.
Authors have stated as overexpression of hsa-miR-193b-3p in hepatocytes could reduce the expression of PPARGC1A mRNA. Have the authors analyzed the PPARGC1A mRNA level when inhibiting hsa-miR-193b-3p expression?
Response: We have not yet analysed PPARGC1A mRNA levels when inhibiting hsa-miR-193b-3p. A more in-depth analysis of the effects of higher or lower levels of hsa-miR-193b-3p is naturally warranted and we plan to do this in the future.
Expression of carbohydrate-responsive element-binding protein (ChREBP) also 181 known as MLX-interacting protein-like, MLXIPL, was reduced by 30% (Figure 3E) when hsa-miR-193b-3p was overexpressed. This transcription factor binds and activates carbohydrate response element (ChoRE). The MLXIPL level was reduced when HSA-miR-193b-3p was overexpressed, which indicates that TG synthesis was inhibited. Then how do hepatocytes increase their lipid content?
Response: The detailed mechanism by which an increase in lipid content is observed is a very interesting question. One hypothesis is that the increased lipid content observed could for example be in part due to the reduced assembly of VLDL given that we also observed lower levels of MTTP.
We also observed a 20% reduction in expression of Fatty Acid Synthase, FASN (Figure 3I) indicating reduced fatty acid synthesis, which was expected since this gene is regulated by ChREBP/MLXIPL. However, no change was observed in Acetyl-CoA carboxylase, ACACA. The same issues were noted when it came to the expression of FAS and ACC, which are both critical enzymes involved in the process of synthesizing fatty acids. The lipid accumulation status in hepatocytes during overexpression of HSA-miR-193b-3p is also controversial.
Response: We agree that the changes we observed in mRNA levels, of genes involved in lipid metabolism, do not provide a clear-cut rationale to explain lipid accumulation in hepatocytes during overexpression of hsa-miR-139b-3p. We have included this comment in the manuscript.
If the authors could analyze hsa-miR-193b-3p's role while it's inhibited, it would be greatly appreciated.
Response: Future work on the subject of the paper will undoubtedly include investigating the effects of inhibiting hsa-miR-193b-3p in hepatocyte. We have included this comment in the discussion.
There is a need to improve the manuscript's language. It is better to avoid using would in several places.
Response: Thank you for the comment and we have revised the manuscript to improve the language.
Reviewer 2 Report
This manuscript provides important findings of miR-193b-3p as diagnostic marker for pre-diabetes. Gene expression of various pathways of glucose and lipid metabolism, glycolysis, gluconeogenesis, lipogenesis, fatty acid synthesis, fatty acid oxidation, TCA cycle and others, in HepG2 cells were measured. All data of gene expression support the hypothesis of using hsa-miR-193b-3p as an early diagnostic biomarker of liver dysmetabolism in pre-diabetes.
Methodology in this study is appropriate, and results are not conflicting.
If, possible, authors should show the criteria for determining pre-diabetes on how high plasma level of hsa-miR-193b-3p.
Author Response
Response to Reviewer 2
This manuscript provides important findings of miR-193b-3p as diagnostic marker for pre-diabetes. Gene expression of various pathways of glucose and lipid metabolism, glycolysis, gluconeogenesis, lipogenesis, fatty acid synthesis, fatty acid oxidation, TCA cycle and others, in HepG2 cells were measured. All data of gene expression support the hypothesis of using hsa-miR-193b-3p as an early diagnostic biomarker of liver dysmetabolism in pre-diabetes. Methodology in this study is appropriate, and results are not conflicting. If, possible, authors should show the criteria for determining pre-diabetes on how high plasma level of hsa-miR-193b-3p.
Response: We welcome comments that enrich the article. The Prediabetes guidelines are based on impaired fasting glucose (IFG: only fasting glycaemia, between 100-125 mg/dl) and/or impaired glucose tolerance (IGT: glycaemia 2h after glucose challenge, between 140-199 mg/dl and with normal fasting glycaemia). We have included this information in the maniscript.
Reviewer 3 Report
In this manuscript, Mollet and Macedo have investigated the consequences of overexpressing a pre-diabetes linked miRNA, miR-193-3p, in HepG2 cells, by measuring the relative expression levels of the predicted target PPAGC1A, intracellular lipid accumulation and mRNA expression levels of genes involved in lipid processing and insulin signalling.
The authors have clearly outlined the research question, rationale and potential long-term prospects of the study.
Can the authors please clarify the following with regards to experimental set up:
1. Do HepG2 cells endogenously express miR-193b-3p? I could not find endogenous miRNA expression mentioned in manuscript. This information is important to be able to understand the effect of O/E in these cells.
2. What is the absolute or relative increase of miR-193-3p in the HepG2 cells in which you have overexpressed miR-193b-3p? Is this increase physiologically relevant for pre-diabetic patients either in liver tissue or plasma? High levels of overexpression can lead to off target effects.
3. For the O/E experiments can the authors explain why a lipofectamine control was used instead of a non-targeting control? A lipofectamine-only control does not control for effects of the increased concentration of miRNA within the cell and specifically on the miRNA machinery.
Fig 2: Microscopic images lacking labels to distinguish between mock and OE193
Author Response
Response to Reviewer 3
In this manuscript, Mollet and Macedo have investigated the consequences of overexpressing a pre-diabetes linked miRNA, miR-193-3p, in HepG2 cells, by measuring the relative expression levels of the predicted target PPAGC1A, intracellular lipid accumulation and mRNA expression levels of genes involved in lipid processing and insulin signalling.
The authors have clearly outlined the research question, rationale and potential long-term prospects of the study.
Thank you in advance for your helpful comments which we have addressed in blue in the responses below.
Can the authors please clarify the following with regards to experimental set up:
- Do HepG2 cells endogenously express miR-193b-3p? I could not find endogenous miRNA expression mentioned in manuscript. This information is important to be able to understand the effect of O/E in these cells.
Response: We did measure the endogenous levels of hsa-miR-193b-3p and have now included this information in the manuscript. HepG2 cells endogenously express hsa-miR-193b-3p. We measured hsa-miR-193b-3p in the HepG2 cells with real-time RT-qPCR using TaqMan® hsa-miR-193b-3p MicroRNA Assay against two endogenous references snoRNAs RNU44 and RNU48.
- What is the absolute or relative increase of miR-193-3p in the HepG2 cells in which you have overexpressed miR-193b-3p? Is this increase physiologically relevant for pre-diabetic patients either in liver tissue or plasma? High levels of overexpression can lead to off target effects.
Response: Using real-time RT-qPCR in HepG2 cells we observed identical Ct values of hsa-miR-193b-3p and the reference snoRNA RNU44 (Ct=26.8±0.4 and Ct=26.7±0.3, respectively), while snoRNA RNU48 had higher expression with Ct=23.7±0.5. Parrizas et al. 2015 showed that higher levels of hsa-miR-193b-3p were observed in plasma of people with impaired fasting glucose by a difference of almost 6 dCt between lowest levels in non-diabetic controls and highest level in impaired fasting glucose, which corresponds to a 64-fold increase. To determine a physiologically relevant overexpression of hsa-miR-193b-3p to use in this study we measured the fold increase in hsa-miR-193b-3p using a 0.05nM, 0.5nM and 5nM transfection of hsa-miR-193b-3p against the scrambled control microRNA and observed a fold increase of 1.9, 11.1 and 131 respectively. We therefore selected 5nM hsa-miR-193b-3p as an appropriate physiologically relevant level of overexpression. This information has now been included in the manuscript.
- For the O/E experiments can the authors explain why a lipofectamine control was used instead of a non-targeting control? A lipofectamine-only control does not control for effects of the increased concentration of miRNA within the cell and specifically on the miRNA machinery.
Response: For the OE we did perform transfections with a negative control scrambled miRNA. This data has now been included in the manuscript and figures. Figures for relative gene expression have been updated to include 4 biological replicates except for SREBF1 and SREBF2 that have 3 biological replicates; technical replicates were all done in duplicate for all. In cases where the scramble control has interfered with the expression of a gene, the discussion focuses on comparison with the mock transfection containing only lipofectamine. Only results with significant differences versus both lipofectamine control and scrambled negative control are mentioned in the abstract.
Fig 2: Microscopic images lacking labels to distinguish between mock and OE193
Response: The nomenclature was in the Figure legend but we have now included the labels in the figure.
Round 2
Reviewer 1 Report
This author has submitted a revised research article entitled Pre-diabetes linked miRNA miR-193b-3p targets PPARGC1A, disrupts metabolic gene expression patterns, and increases lipid accumulation in hepatocytes: relevance to MAFLD. In response to the reviewer's comments, the authors have substantially revised their submitted research article. The manuscript has been significantly improved over the original submission, containing all the necessary information and hypotheses. Additionally, the language of the manuscript has been improved. Hence I recommend that this article be accepted in its current form.